# Perceptions of the Use of Alcohol and Drugs after Sudden Bereavement by Unnatural Causes: Analysis of Online Qualitative Data

**DOI:** 10.3390/ijerph17030677

**Published:** 2020-01-21

**Authors:** Lauren Drabwell, Jessica Eng, Fiona Stevenson, Michael King, David Osborn, Alexandra Pitman

**Affiliations:** 1UCL Centre for Behaviour Change, Research Department of Clinical, Educational, and Health Psychology, University College London, 1-19 Torrington Place, London WC1E 7HB, UK; lauren.drabwell.17@alumni.ucl.ac.uk; 2UCL Division of Psychology and Language Sciences, 26 Bedford Way, London WC1H 0AP, UK; j.eng.17@alumni.ucl.ac.uk; 3UCL Research Department of Primary Care & Population Health, Rowland Hill St, London NW3 2PF, UK; f.stevenson@ucl.ac.uk; 4UCL Division of Psychiatry, Maple House, 149 Tottenham Court Road, London W1T 7NF, UK; d.osborn@ucl.ac.uk (D.O.); michael.king@ucl.ac.uk (M.K.); 5Camden and Islington NHS Foundation Trust, St Pancras Hospital, London NW1 0PE, UK

**Keywords:** bereavement, grief, accidental death, alcohol, drugs, qualitative research, thematic analysis

## Abstract

Bereavement is associated with an increased risk of psychiatric morbidity and all-cause mortality, particularly in younger people and after unnatural deaths. Substance misuse is implicated but little research has investigated patterns of drug or alcohol use after bereavement. We used a national online survey to collect qualitative data describing whether and how substance use changes after sudden bereavement. We conducted thematic analysis of free-text responses to a question probing use of alcohol and drugs after the sudden unnatural (non-suicide) death of a family member or a close friend. We analysed data from 243 adults in British Higher Education Institutions aged 18–40, identifying two main themes describing post-bereavement alcohol or drug use: (1) sense of control over use of drugs or alcohol (loss of control versus self-discipline), (2) harnessing the specific effects of drugs or alcohol. Across themes we identified age patterning in relation to substance misuse as a form of rebellion among those bereaved in childhood, and gender patterning in relation to men using alcohol to help express their emotions. The limitations of our sampling mean that these findings may not be generalizable from highly-educated settings to young people in the general population. Our findings describe how some young bereaved adults use drugs and alcohol to help them cope with traumatic loss, and suggest how clinicians might respond to any difficulties controlling substance use.

## 1. Introduction

Bereavement is a near universal experience, and grief is a normal response to loss that most people adjust to without psychological intervention [1]. Theories of bereavement help describe the processes of grieving involved in adjusting to the loss, and include those developed by Freud [2], Bowlby [3,4], Kubler-Ross [5], Worden [6], Murray Parkes [7], Stroebe and Schutt [8], and Rubin [9]. Whilst some theories describe phases [3,4], stages [5], or tasks [6] of mourning that the bereaved are expected to move through, there is also an acknowledgement that individuals can move backwards and forwards between these [6,8] in a dynamic process. More recent theorists regard bereavement as never fully resolved, but as a way of retaining the memory of the deceased [10]. In the dual process of grieving, Stroebe and Shutt describe bereaved individuals oscillating between loss orientation, in which they focus on their grief and pain, and restoration orientation, in which they distract themselves from grief in order to cope [8]. Rubin’s two-track model of bereavement considers biopsychosocial functioning as a separate dimension to the bereaved person’s attachment to the deceased [9], with the bereaved addressing both of these. These theoretical concepts help map what might be regarded as ‘normal’ grieving, and where individuals might struggle with specific tasks, stages, or adjustments to functioning.

Although grieving is a natural process by which people adapt to loss, bereavement is associated with an excess risk of physical and psychiatric morbidity and all-cause mortality [1,11,12,13], and these risks are influenced by the nature of the loss [14]. Violent deaths are understood to be particularly traumatic for the bereaved, and are associated with the development of post-traumatic stress disorder and depressive disorders [14], for which the trajectory of recovery may be slower than after other bereavements [14]. The mortality risks following bereavement are particularly high in the first six months after the loss, and where the death was due to accidental or violent causes or alcohol-related diseases [1,15]. The risk of suicide is also elevated in people who experience bereavement, particularly in those bereaved by suicide, homicide or accidental death [16]. Other sequelae of bereavement include prolonged grief disorder, recently included in psychiatric diagnostic classification systems [17], which recognizes that some people struggle to transition from acute to integrated grief. This diagnostic entity arises from descriptions of complicated grief, in which prolonged, unresolved and intense grief is associated with substantial psychosocial functioning, and prolonged grief disorder, in which core symptoms include yearning for the deceased and difficulties in psychosocial functioning beyond 6 months post-loss [18]. The risk of complicated grief is increased after sudden or violent losses [14,19]. The public health impact of bereavement is therefore substantial, and identifying explanations for these excess risks of morbidity and mortality is important so that we address contributing factors.

The developmental stage at which a loss occurs is also important. Bereavement at a young age is particularly challenging because, being uncommon, it sets the bereaved apart from their peer group, who may lack the understanding to support them appropriately. The death of a parent may leave a young person to be cared for by a grieving surviving parent. These deficits in the informal support network are concerning given the observed reluctance of young people to seek professional help for mental health problems [20]. US research shows that young people who lose a sibling or close friend are more likely to develop complicated grief than their non-bereaved peers, and to report significantly higher levels of depression, and somatic symptoms [21]. Studies using Danish registers show that the risk of self-harm, psychiatric illness and suicide amongst the bereaved is greatest in younger age groups [16], and that experiencing the death of one or both biological parents significantly increases the risk of suicide attempts in young people [22]. Studies using Swedish registers also demonstrate an increased risk of self-harm in children bereaved by the death of a sibling [23], and of psychiatric treatment in children bereaved by the death of a parent [24]. Across all age groups, mental health and mortality are similar in those bereaved by suicide and by accidental deaths [25], suggesting that it is the violence or unnatural nature of the loss that influences outcomes. The stigma associated with violent or unnatural deaths may also limit support available [26].

One explanation for the increased risk of physical and psychiatric morbidity and all-cause mortality after bereavement is thought to be increased alcohol use following the loss [1], as evidenced in Finland by the greater risk of dying from alcohol-related diseases amongst the bereaved [27]. In a United States sample, 4% adolescents bereaved by the death of a parent had an alcohol or drug use disorder, compared to 0% of non-bereaved controls, and the bereaved also had an excess risk of depression [28]. Alcohol and drug misuse problems are risk factors for physical health problems and mortality (whether due to natural and unnatural causes) [29] and there is an established link between alcohol and drug misuse and suicide risk [30]. There is therefore great interest in understanding the role of alcohol in psychiatric morbidity and all-cause mortality after bereavement. Studies using self-report measures identify increases in substance use after bereavement among those with [31] and without [32] pre-existing substance misuse problems. Studies using population registers find a similar risk of drug or alcohol misuse among people bereaved by suicide and by other unnatural causes [33]. However, whilst these offer opportunities to conduct the most methodologically rigorous longitudinal population-based studies, they only include data on diagnosed substance misuse disorders and not specific patterns of drinking or drug use. Given the range of drinking and drug use patterns observed in practice, it is more informative to study patterns of harmful and hazardous substance use. These are better described using anonymous surveys and qualitative methods. Our aim in this study was to investigate how young adults bereaved by sudden unnatural death of a close friend or relative describe their drug and alcohol use after the loss. Our focus on those aged 18–40 years defined a group who are of particular policy interest in relation to mental health, substance misuse, and help-seeking. We hoped that this work would help understand health behaviour in this age group, and the development of more appropriate support for young people who experience traumatic bereavement.

## 2. Materials and Methods 

### 2.1. Methodological Approach

Our research question was whether people bereaved suddenly by unnatural causes perceive any changes in their use of alcohol or drugs following the death, and if so, what the nature of such changes were, and explanations respondents provided for the motivations behind such changes. Given that all the researchers on the team were based in university health departments, we acknowledged our pre-existing beliefs and attitudes towards drug and alcohol use in relation to their risks and benefits. We conducted a phenomenological study, and took an inductive approach, using thematic analysis.

### 2.2. Participants and Design 

We collected data as part of the UCL Bereavement Study, the sampling methods for which are described in detail elsewhere [34]. In brief, in 2010, we invited 164 UK Higher Education Institutions (HEIs) to take part in an online cross-sectional national survey, to explore the impact of sudden bereavement. Of these, 37/164 HEIs agreed to participate in the survey by emailing the survey link in individual email invitations to all students and staff. Ten of these HEIs advertised the survey in their weekly email newsletters, on their student and staff intranets or by sending the link to students only, due to the sensitivities of the topic. Participating HEIs provided a broad sample of student and staff bodies across England, Wales, Scotland and Northern Ireland, balancing agricultural, academic and arts institutions. 

Our inclusion criteria were adults aged 18-40 years who had, since the age of 10, experienced the sudden bereavement of a close contact, who we defined as “a relative or friend who mattered to you, and from whom you were able to obtain support, either emotional or practical”. The eligibility criteria excluded those who had experienced bereavement in childhood, in order to reduce recall bias. The email inviting participation contained a link to an online questionnaire, which consisted of 119 closed questions on participants’ socio-demographic and clinical characteristics and 20 open questions addressing specific aspects of bereavement. These covered domains such as the impact of the bereavement on finances, educational and work functioning, and relationships. The questionnaire was designed by the senior authors in collaboration with a consultation group of young bereaved adults and bereavement counsellors (see Appendix A), and piloted with the clients of four national bereavement support organisations. The questions were carefully worded to be neutral and non-leading, to avoid distress and the assumption of solely negative outcomes after bereavement. We did not set a word limit on free text answers. Participants were requested to provide as much or as little information as they wanted, or to skip the question if they wished to. On completion of the survey, all participants were directed to a list of support services at their HEI.

Among the 20 open questions we asked one question about the impact of the bereavement on use of drugs or alcohol, as follows: ‘In what way, if any, has the bereavement affected your drinking habits or your use of unprescribed drugs?’ (Unprescribed drugs include illicit drugs as well as medications used above their prescribed limits.)” To avoid ambiguity we used the term unprescribed drugs to cover use of illegal drugs, legal highs, over-the-counter drugs, or prescribed drugs used above advised limits. 

Of those who received the email and met eligibility criteria, 3432 respondents consented to participate and completed the questionnaire. In the current study we analysed data on a nested sample of participants who self-reported exposure to bereavement by unnatural causes, including due to road crashes, homicide, and industrial accidents, but excluding suicide. 

### 2.3. Ethics

We gained ethics approval for the study protocol from the UCL Research Ethics Committee in 2010. All participants provided informed consent online. The survey participation information leaflet stated that all data would be handled in accordance with data protection legislation and anonymised to protect the identities of participants.

### 2.4. Analytic Approach

We conducted thematic analysis of responses to the survey question [35] on use of drugs or alcohol. First we imported responses from the secure survey platform into Microsoft Excel, with which we were able to support the organisation and coding of large volumes of relatively short responses. Our analytic team covered the disciplines of behaviour change science (Lauren Drabwell), psychological sciences (Jessica Eng), and academic psychiatry (Alexandra Pitman), bringing a range of perspectives. We took a transdiagnostic approach, focusing not on whether respondents’ behaviour mapped to concepts of dependence or misuse, but on the meaning of their relationship to drinking or drug use. Our team meetings encouraged challenges to coding decisions to promote discussion and improve reflexivity.

In the first stage of the analysis we removed ambiguous responses, including those indicating ‘not applicable’, such that only responses with interpretable meaning were analysed. This enabled familiarisation with the data. The first two authors then proceeded to a deeper exploration of the meaning of participants’ experiences, by independently coding the first 100 responses in the dataset to generate initial codes. Having compared initial codes to review consistency between coders as a check on robustness, and agreed an initial coding framework within the context of wider group discussions, the first author then continued to code the full dataset, building up a framework of new codes, sub-codes, and collapsed codes. Themes were identified in collaboration with the other two researchers in cycles of analysis. Regular discussion meetings within the research team reviewed themes, and encouraged reflexivity and enhanced the validity of our findings by providing opportunities to challenge others’ assumptions and refine our interpretations and analytic processes. Senior authors brought perspectives from the disciplines of medical sociology (Fiona Stevenson) and academic psychiatry (Michael King, David Osborn). One aspect of our discussions was our interpretation of the degree to which respondents appeared to feel that that use of substances was harmful, and the extent to which an awareness of harmfulness was balanced with any perceived benefits.

Finally, the researchers reviewed sub-codes against higher-order themes to validate the coding framework and ensure conceptual coherence, agreeing on the final definition and naming of themes.

We followed COnsolidated criteria for REporting Qualitative Research (COREQ) guidelines in reporting our findings [35]. We presented our themes with illustrative quotes, making minor spelling corrections where letters were inverted, or omitted. 

## 3. Results

### 3.1. Response and Participant Characteristics

Of the 3432 respondents who completed the UCL Bereavement Study questionnaire, 742 participants self-identified as having experienced bereavement by unnatural non-suicide causes, and of these, 363 answered the question on drug and alcohol use. We excluded responses in which their response indicated “not applicable” or equivalent (*n* = 120).

The majority of our sample of 243 adults were female (80%), and 72% of respondents were full-time students (Table 1). 

Alcohol use was described more commonly than drug use, and in the latter case primarily related to cannabis use. It was common for respondents to report a change in drug or alcohol use. Our thematic analysis identified two main themes, and Table 2 summarises these themes and sub-themes, detailed further below with illustrative quotes. 

### 3.2. Thematic Analysis

#### 3.2.1. Theme 1: Sense of Control over Use of Drugs or Alcohol

Many of the individuals’ views on their changes in drug or alcohol use were defined in relation to their sense of control over these behaviours. Their perceptions of excessive substance use were primarily expressed in negative terms or in relation to negative consequences:

(1) Loss of control of substance use. It was common for respondents to describe a loss of control over their substance use, especially alcohol. They viewed this primarily as undesirable, describing excessive use, often with a binge pattern. For many the period immediately following the bereavement was characterised by excessive use of drugs or alcohol, with this spike in use tapering off over time, either as the bereavement was processed and/or with a growing awareness of the potential harms:


*“For about a month I started binge drinking, as well as upping my dose of antidepressants without medical advice” (21 year old female, 1 year since uncle/aunt died)*



*“In the first month following his death I was drinking excessively and smoking marijuana a lot” (22 year old male, less than a year since death of close friend)*


Beyond a transient period of excessive use, some respondents described an awareness of a transition to dependence on substances, albeit not always permanent:


*“For 2 years relied on alcohol and cannabis daily....” (31 year old female, 7 years since death of close friend)*



*“Became an alcoholic. Used drink every day for more than a year after bereavement ....” (20 year old female, 2 years since death of close friend)*


For some individuals their loss of control over alcohol or drugs had become apparent to them in their dysfunctional consequences, whether merely embarrassing or dangerous. This was particularly alarming for individuals where it had led them into risky situations, displaying an awareness of the potential harms: 


*“After she died I drank so heavily I wet my bed on several occasions” (25 year old female, 4 years since death of close friend)*



*“We started drinking a lot more after it happened and sometimes I got myself into dangerous situations because of it.” (19 year old female, 17 years old since close friend died)*


(2) Self-discipline over substance use. The bereaved often framed their alcohol and drug consumption with reference to their efforts to control the substances they used, explaining that they limited their use due to the fear of negative consequences. This ranged from avoiding the substance entirely as a means of self-protection, to reducing their use to a level perceived to lie within safe limits.


*(i) Avoidance of drugs or alcohol for self-protection*


It was common for individuals to report deliberately avoiding specific substances, particularly alcohol, to avoid intoxication bringing up negative emotions. Avoidance was therefore seen as a form of self-protection, whether from the mood-lowering effects of specific substances or to contain emotions. Participants referenced a fear of becoming depressed as a disincentive to drinking, either arising from the depressant effects of alcohol or because of an awareness of their own tendency to react badly to it: 


*“I was afraid to drink too much as I know alcohol is a depressant (I was depressed enough!)” (26 year old female, 6 years since death of ex-partner)*



*“Then I found that alcohol made me very depressed so I stopped drinking completely” (29 year old female, 8 years since death of mother)*


There was a striking sense of fear around becoming emotional, seemingly relating to a dread of losing emotional control. Emotional outbursts were either feared due to the embarrassment of becoming tearful in the company of others, or because of the emotional burden itself:


*“Tended to avoid drinking as scared of losing control of my emotions….” (21 year old female, less than a year since death of close friend)*



*“Since he died I have chosen not to get stoned again because I am concerned for the mood effects of smoking weed and I am scared of being very sad” (27 year old female, 6 years since death of father)*


This sub-theme contrasted with the sub-theme below regarding use of substances to handle emotions, and specifically the use of substances to achieve emotional openness.


*(ii) Avoidance due to association of the death with substance use*


Specific experiences of sudden deaths attributed to drug or alcohol use appeared to have influenced the behaviour of the bereaved. This ranged from a sense of wariness around substance use as a means of respecting the deceased, to a specific avoidance of certain substances or certain risk-taking behaviour: 


*“I drink less, and am more aware of people who have been drinking and am more cautious” (26 year old female, 2 years since death of close friend)*



*“If I am driving and I will not get into a car with someone who has been drinking that day.” (23 year old female, 2 years since death of close colleague)*


Where alcohol or drug addiction was specifically implicated in the death, individuals described a determination to avoid ending up the same way. This sense of wariness and caution, and an awareness of the dangers, appeared to have heavily influenced their drug and alcohol consumption:


*“Due to the nature of this person’s death being a drug overdose, the bereavement helped me to use drugs a lot less than before the bereavement, until eventually, I stopped using them all together.” (22 year old female, 4 years since death of close friend)*



*“I am scared by the thought of becoming an alcoholic like my aunt so I have been more careful with drink and have completely avoided drugs.” (30 year old female, 8 years since death of aunt)*


#### 3.2.2. Theme 2: Harnessing the Specific Effects of Drugs or Alcohol

Many of the participants who described an increase in their drug and alcohol use reported that this was a conscious decision, to achieve a specific aim. This ranged from drinking with others to honour the memory of the deceased, to using drugs or alcohol to self-medicate. This theme intersected with theme one in that it described the reasons for increased use, whether controlled or not, conveying a sense of whether the benefits were worth the potential harms:

(1) In memoriam. Some individuals changed their drug and alcohol consumption as a way of honouring or memorialising the deceased and the way they had lived their life. This sub-theme lay in direct contrast to the sub-theme under Theme 1 describing avoidance of drugs or alcohol due to their associations with the death itself. Those who chose to memorialise the deceased with alcohol or drugs described doing so with others, reflecting a means of gaining social connectedness with other bereaved people.


*“Initially after his death, drank more with friends at many celebrations of his life.” (24 year old female, less than a year since death of close friend)*



*“When I am with (J’s) friends I can drink and feel a part of J is there too. I drink more around them.” (22 year old female, 1 year since death of partner)*


For a small number of respondents, their use of drugs or alcohol reflected a desire to emulate the deceased or to maintain the presence of the deceased, with the sense that modelling their behaviour provided a direct connection to them:


*“Drink more on nights out as she was a party animal!” (21 year old female, 3 years since death of close friend)*



*“My deceased friend had taken a lot of drugs and I found it a way to connect in some weird way” (32 year old female, 11 years since death of close friend)*


(2) Substance use as a release. The ability to lose oneself in an intoxicated state was valued by some respondents as a means of escaping. Such states provided brief episodes of pleasure, or a shift in mind set towards an awareness of a need to gain the most out of what might be a brief life. 


*(i) To experience pleasure*


Individuals referenced their drinking and drug taking as a means of providing a sense of pleasure during a difficult period. Such intense periods of pleasure appeared to provide distraction and a connection with others:


*“Still smoke as much weed as ever, and clearly had no intention of stopping after the bereavement. Pretty much the only thing that brought me a bit of joy at the time.” (19 year old female, 2 years since death of father)*



*“I took drugs more often and was able to experience a lot of happiness with friends in this time” (22 year old female, 5 years since death of father)*


The psychoactive effects of drugs appeared to provide a means of processing the loss whilst also escaping from its pain. Some related the pleasure gained through psychoactive drugs as derived from their ability to understand themselves better, linking to the below sub-theme on use of substances to cope with emotions.


*“I definitely had periods of ’escapism’ proceeding the death. …. The death did perhaps put me in touch with very deep levels of abstract introspection, which, when made conscious and manipulated by the effects of psychoactive drugs, particularly hallucinogens, aided my sense of existentialism to wonderful heights.” (23 year old male, 6 years since death of close friend)*



*(ii) To live life to the full*


In a small number of cases, individuals indicated they had shifted in their attitudes to life, and sought to consume more substances, both from a sense of gaining more varied life experiences and also in living life to the full: 


*“I think that I felt that taking drugs after the bereavement was about ’living life’ and not wasting time...” (32 year old female, 11 years since death of close friend)*



*“I feel as though I may drink more as a result as I have developed a different attitude to life. I am more of a live in the moment type of person now and try to make sure I always enjoy myself.” (22 year old female, 6 years since death of close friend)*



*(iii) To sleep*


Some individuals would consume alcohol, illicit drugs and prescription drugs in order to sleep. Again, this and the sub-theme below, suggested that substances were being used to achieve an escape from uncomfortable emotions or from ruminating about the death, which might otherwise trouble them during the period before sleep:


*“I use strong painkillers as a sleeping aid, as over the counter drugs don’t work as well” (23 year old female, 2 years since death of close friend)*



*(iv)To escape reality*


Some individuals chose to engage with substances as a form of distraction from the pain of their loss. This was in direct contrast to those who used drugs or alcohol to process their thoughts, and instead reflected an avoidance of thinking about the bereavement, by means of distraction:


*“Straight after the accident, I didn’t care and I drank heavily and I went to Amsterdam a lot so I could sleep and distract my mind with the use of drugs.” (36 year old female, 10 years since death of partner)*



*“After the death I smoked a bit of cannabis to alleviate the pain and find some distraction, to stop having obsessive thoughts.” (38 year old female, 2 years since death of close friend)*


Some people described substance use to escape reality, reflecting a desire to forget painful memories temporarily until they were ready to process them: 


*“I drink a lot and have taken unprescribed drugs …after (my sister) died I drank every day for about a year. I couldn’t face reality” (27 year old female, 5 years since death of sister)*



*“I drank a lot after he died and if I think about him now and I am out drinking with my friends I will drink more to try and forget.” (21 year old female, 5 years since death of partner)*


(3) Use of substances to cope with emotions. Respondents described using drugs or alcohol to help them cope with the extreme emotions they were processing after the loss:


*“I did use painkillers more readily, those designed to be used for migraines, but instead of me using them when I had a migraine I would use them when I couldn’t cope.” (27 year old female, 3 years since death of father)*



*“In the months following my father’s death I drank a lot more…I remain, to this day, pleased that this was my course of action. It helped me to deal with the loss slowly, in step…” (22 year old female, 5 years since death of father)*


The language used indicated a need to relax and to numb emotions, self-medicating the physical and mental strain of their grief in a manner that implied a degree of control over use: 


*“I have started to smoke marijuana again, as this calms me down” (20 year old male, less than a year since death of cousin)*



*“For a period I used to drink to numb the pain but no longer do this” (40 year old female, 23 years old since death of partner)*


Drinking or taking drugs to cope with emotions was not always effective, and some eventually recognised this as being an unhelpful coping mechanism:


*“I think I would sometimes drink to try and forget about it but it never helped” (21 year old female, 2 years since death of close friend)*



*“If I drink I drink a lot because I think it will make me better, but it makes me worse” (20 year old female, less than a year since death of cousin)*



*(i) Substance use to express frustration*


A few people described substance use as a way of expressing their anger at the loss of a close friend or relative. This was more common among those who had experienced bereavement as a child, perhaps reflecting their greater difficulties articulating their grief and the lack of shared experience among their peers. This was described as a form of rebellion, but was usually transient and was recognised as being a dysfunctional way of coping with the bereavement: 


*“I began smoking as a form of rebellion, anger and frustration. I found it hard to comprehend why such a nice young person would need to be taken away and I would wish that it had been me and not her.” (18 year old female, 4 years since death of death of close friend)*



*“I was not old enough at the time to drink. I tried illicit drugs earlier than most. Possibly an act of rebellion which didn’t last” (21 year old female, 8 years since death of brother)*



*(ii) Substance use to achieve emotional openness*


A few respondents described alcohol (and to a lesser extent drugs) as beneficial in enabling them to open up to others about their grief. Despite the predominance of females in our sample, this sub-theme was more apparent among men, suggesting that alcohol or drugs provided a means of overcoming barriers to help-seeking: 


*‘temporarily found drinking helped me open up to myself and others more so drinking increased temporarily…’ (19 year old male, less than a year since death of close friend)*



*‘I’d also get drunk at parties where I didn’t know many people and cry and talk to people about (it)’ (19 year old female, 2 years since death of close friend)*


This sub-theme also contrasted with the sub-theme Avoidance of drugs or alcohol for self-protection, in which individuals (primarily female) avoided drugs or alcohol to protect themselves from revealing the extent of their emotions to others.

## 4. Discussion

### 4.1. Main Findings

This study identified two main themes describing the experiences of people bereaved by unnatural causes in relation to perceived changes in their use of drugs or alcohol and their motivations for this. These themes were a sense of control over use of drugs or alcohol, and efforts to harness the specific effects of drugs or alcohol. In this first theme a number of respondents described struggles in controlling their substance use, with many reporting a spike in drinking or drug use immediately after the loss, tailing off in subsequent months in favour of alternative coping mechanisms. Others had a more rigid control over their use of substances, some fearing the negative consequences that they associated directly or indirectly with the cause of their loved one’s death. In the second theme respondents described taking control over specific substances to derive beneficial effects, whether as a means of release or rebellion or to help express emotions and connect with others. Such efforts were not always effective, or came with an awareness of potential harms. These themes therefore captured a tension between two key dimensions: an awareness of the utility of drug or alcohol use (for example in experiencing pleasure, achieving an escape, or expressing frustration), as seen in theme 2, and an awareness for some that this was at the cost of difficulties controlling use, as captured in theme 1, sometimes with harmful consequences.

These qualitative findings illustrate the ways in which people who experience traumatic bereavement use drugs or alcohol, or for some avoid drugs or alcohol, as a coping strategy. There were clear examples of self-medicating to palliate specific aspects of grief, and a more general need for release and escapism. However, some individuals came to recognise that substance use to escape grief was unhelpful for them, particularly where losing control had led to embarrassing or risky consequences. We identified age patterning of themes only in relation to use of drugs or alcohol as a form of rebellion, which was more apparent amongst those bereaved in childhood, perhaps reflecting their cognitive stage of development. We also identified gender patterning of themes in relation to the tendency of men to use alcohol to help express their emotions to others. We did not find any patterning regarding kinship, as there were no apparent differences between those who had lost a relative and those who had lost a friend.

### 4.2. Results in the Context of Other Studies

Only one other study, to our knowledge, has specifically sought to use qualitative methods to explore substance use patterns of people bereaved through unnatural causes. This was our own analysis of responses to UCL Bereavement Study data restricted to people bereaved by suicide [36]. Comparing the two studies, we find many similarities, which is unsurprising given the features of trauma and violence common to both bereavement experiences and their similarities in terms of mental health outcomes [37]. The two main themes we identified in the current study mapped to two of the three themes identified in the study describing the suicide-bereaved [36]. In both samples, therefore, struggles over control of drugs or alcohol were prominent, as were the use of substances for specific gains. In both studies many participants were aware of the harmful consequences, and a transient spike in substance misuse was apparent in both samples. The key differences between the findings of the two studies were that in our sample of adults bereaved by non-suicide sudden loss it was apparent that some, particularly men, used alcohol to help express their emotions when communicating with others. This was not apparent in our analysis of data for the suicide-bereaved, who seemed to use drugs or alcohol to block out or escape painful emotions, or cut out alcohol to avoid becoming overwhelmed by emotions. Whilst the sub-theme of escapism was identified in the current study, this was couched more in terms of a release, and less explicitly as a means of avoiding experiencing emotions. 

Our findings of a male tendency to use alcohol to help discuss feelings is consistent with the findings of a qualitative study exploring the experience of young Irish men recently bereaved by the suicide of a male friend [38]. Interviewees described feeling reluctant to discuss their personal problems with other men, and found that when drinking together they were better able to disclose their concerns and feelings. These findings are consistent with a US study finding that women are more likely to open up to others about their grief [20], and with theoretical work describing gender differences in cognitive processes [21] and a distinct pattern of male grieving that includes alcohol use [39].

The findings of the current study suggest that excessive use of alcohol or drugs was a problem for many of our sample in the immediate aftermath of the death but tended not to persist longer-term, although we were not able to measure the quantities of substances consumed. This finding is not consistent with quantitative findings from a nationally representative Hungarian sample of adults bereaved by any cause [40]. This found alcohol use to be significantly higher in bereaved men compared to non-bereaved controls at two years after the loss, but not at one year or three years. There were no differences for women. This suggests that the spike phenomenon we observed in our two studies’ samples may be specific to those bereaved by unnatural causes. Our findings regarding use of substances for specific purposes matches those of a study of Swedish widows, who reported using increased amounts of sedatives and alcohol to medicate their grief [32].

The sub-themes we identified also map to different bereavement theories. For individuals who described themselves using substances to escape reality it could be said that they were at Bowlby’s phase of numbing [3,4], or at Kubler-Ross’s stage of denial [5]. In line with Strobe and Schutt’s dual process model, they could also be said to be coping by distracting themselves from loss orientation through oscillating towards restoration [8]. For those who were using alcohol to cope with emotions it could be said that they were struggling with Worden’s task of working through the pain and grief [6]. The spike in substance use described by some could represent a way of coping with the phase of numbing or denial, or a means of distraction. Those who used substances to memorialise the deceased could be said to have found a way to adjust to the environment in which the deceased was missing [3,4,6]. Thus all the above might be considered ‘normal’ aspects of grief. However, participants who described longer-term struggles to control their substance use might be regarded as stuck at a phase of numbing or denial, causing problems with their psychosocial functioning [9].

### 4.3. Strengths and Limitations of Study

This is one of only two British studies to explore specifically the experiences of adults bereaved by sudden unnatural deaths in relation to drug and alcohol consumption, probing patterns of behaviour with qualitative methods to capture information that goes beyond solely diagnostic categories. Our study addresses a gap in research on the determinants of health outcomes after unnatural death, and has clear implications for clinicians involved in the care of young people. Our sampling methods accessed a large sample, representative of those working or studying in higher education in the UK, and not solely the views of those seeking help for substance misuse or bereavement. Our online survey was well suited to collecting sensitive data on substance use in that anonymity promoted disclosure. However, this was balanced against the lack of opportunity to probe meaning, inquire about the precise substances used or quantities consumed. Despite the advantages of our sampling methods, they also created the potential for non-response bias and selection bias [41]. Our sample is therefore more representative of white, highly-educated females from more affluent socio-economic backgrounds than of young people in the general population. The culture of drinking and drug-taking among British students [42] ,perpetuated by social factors [43], may also differ from that in students in other countries [44]. Students in the UK or Ireland are more likely to abuse drugs [45] or alcohol [46] than their non-student peers. Our collaborative team approach to analysis and discussing reflexivity improved the validity of the final thematic framework, and the range of academic and health backgrounds encouraged challenges to each researcher’s interpretation of the data. In analysing data separately for people bereaved by suicide [36] we were able to compare their accounts with those of people bereaved by sudden accidental death, and identify where themes were similar and any unique aspects. Although our data were collected in 2010, the findings of this study remain valid because Global Burden of Disease data for the period 1990 to 2013 show that alcohol remains the leading risk factor for ill-health, early mortality and disability among people aged 15 to 49 years [47]. Additionally, as this is the only study to have investigated this specific research question, these data constitute the best available evidence. 

### 4.4. Clinical, Policy, and Research Implications

Our findings suggest that it is common for young adults to report a change in drug or alcohol use after traumatic bereavement, whether by increasing their use to self-medicate aspects of grief, or restricting use as an alternative coping strategy. The former group described binge drinking, an awareness of addiction, and a range of harmful consequences to their increased use. The potential for negative health outcomes is apparent, and this analysis provides a means for clinicians to understand the context for this health behaviour in young bereaved adults, and how best they might intervene. A transient spike in drinking or drug-taking was observed after the loss, much as it was in our study of young adults bereaved by suicide. This suggests that for some young bereaved people a watchful approach may be more acceptable; monitoring drug or alcohol misuse to ascertain whether it is self-limiting or if sensitive interventions are required to address emergent addiction. This more nuanced understanding helps clinicians move away from a paternalistic, punitive approach to substance use and understand where it might be a temporary phenomenon, used for a specific aim, and where brief alcohol interventions or referral to substance misuse services are indicated. Similarly young bereaved people who restrict their use may require monitoring to ensure that their thoughts do not become over-valued, leading to rumination and anxiety. Our work is also of interest to bereaved individuals, who may recognise the experiences described and the general awareness of harmful effects, which might motivate them to seek help for excessive substance use.

From a policy perspective it is clear that young people bereaved by unnatural causes struggle after the loss, with many using drugs or alcohol as a coping mechanism, and that access to psychological support for this group could be improved. There is not strong evidence for motivational interviewing [48] or social norms interventions [49] to prevent alcohol misuse among young adults. However, given the specific reasons for using substances as described in this study, tailored alcohol interventions may be needed for young bereaved adults. Counselling services in higher educational settings sometimes lack expertise to support the bereaved, but may be the only source of support to new students who lack a robust social support network. Alcohol services may also lack expertise to support the bereaved, but might benefit from an awareness of the issues described here. Regardless of where support is best situated, accessing such support is likely to be influenced by the stigma of sudden bereavement and of substance misuse. Research shows that people bereaved by unnatural causes feel more stigmatised than those bereaved by natural causes [50], and that this stigma is implicated in adverse mental health outcomes [34]. The stigma of sudden bereavement is itself a risk factor for suicide attempt [51], with findings from qualitative [52] and quantitative [26] research suggest that this stigma can limit support from others [50]. By marketing support services to encourage young people bereaved by unnatural causes to access support for substance misuse and other difficulties, this stigma would be challenged whilst also encouraging help-seeking.

Regarding future research, there is much more to understand about patterns of substance use after bereavement, particularly in ethnic minority groups and sexual minority groups. Cohort studies that measured drug and alcohol use, perhaps using ecological momentary assessment (EMA) would help understand the triggers for substance use after bereavement, helping explore how best to intervene to break cycles of harmful use. Further qualitative work would help explore with young people what support they would find acceptable, whether intervening early after a loss would be perceived as too intrusive, and how they would advise their peer groups to support them.

## 5. Conclusions

Our thematic analysis of qualitative online survey data describing the drug and alcohol use of young people bereaved by unnatural causes of death identified that it was common for this group to struggle in controlling their use, and to use drugs or alcohol to cope with their emotions. Our two main themes described a sense of control over use of drugs or alcohol, and harnessing the specific effects of drugs or alcohol. These findings are of great value to clinicians and bereavement counsellors, providing a more nuanced understanding of the challenges faced by young bereaved people, who may lack informal support from their peer groups. In understanding the motivations behind drug or alcohol use in young people after traumatic loss, and their social consequences, we will be better able to develop the support sources appropriate to their needs.

## Figures and Tables

**Table 1 ijerph-17-00677-t001:** Participants’ Sociodemographic Characteristics.

Characteristic		Total *n* = 243
Gender		*n* (%)
	male	48 (20)
	female	195 (80)
Age		
	18–21	87 (36)
	22–25	69 (28)
	26–30	40 (17)
	31–40	47 (19)
Ethnicity		
	White	228 (94)
	Non-White	15 (6)
Work status		
	full-time student	175 (72)
	part-time student	10 (4)
	full-time job	35 (14)
	part-time job	11 (5)
	other	12 (5)

**Table 2 ijerph-17-00677-t002:** Table describing themes and sub-themes.

	Main Themes	Sub Themes
1	Sense of control over use of drugs or alcohol	Loss of control of substance use Self-discipline over substance use ○ avoidance of drugs or alcohol for self-protection ○ avoidance due to association of the death with substance use
2	Harnessing the specific effects of drugs or alcohol	In memoriam Substance use as a release ○ to experience pleasure ○ to live life to the full ○ to sleep ○ to escape reality Use of substances to cope with emotions ○ substance use to express frustration ○ substance use to achieve emotional openness

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
