# Peer review of "Perceptions of the Use of Alcohol and Drugs after Sudden Bereavement by Unnatural Causes: Analysis of Online Qualitative Data"

_ijerph, 2020, doi:10.3390/ijerph17030677_

Round 1

Reviewer 1 Report

Perceptions of the use of alcohol and drugs after sudden bereavement by unnatural causes: analysis of online qualitative data

This wide-ranging qualitative study examined texts written by 363 adolescents after the death of someone close to them. This study is very important and can help in understanding the consequences of bereavement on risky behaviors such as alcohol and drug use. The authors need to reinforce the theoretical anchor of the research, formulate the qualitative research procedure such that it will be clear to the reader and provide a more in-depth analysis of the data.

Therefore, I recommend accepting the paper after the following changes have been made:

Introduction

Some references should be replaced with more recent literature. For example: #10 The authors did not define complicated grief. What percentage of the bereaved adolescents use alcohol and drugs in comparison with the general (non-bereaved) adolescent population? The introduction surveys research in this field but lacks a theoretical anchor. Many theories have been proposed for coping with grief and loss. I recommend the two-track bereavement theory proposed by Prof. Rubin.

Materials and Methods

The authors need to indicate what type of qualitative research they conducted and why: phenomenological? narrative? ethnographic? The data were collected in 2010. Please add an explanation of the current relevance of the analysis or indicate the research limitations. The writers put together the questionnaire. What sort of validation process was used? Was the questionnaire tested in a pre-test?

Appendix S1 wasn’t attached.

Further information about the 'reflection process' (page 4) would make this section more meaningful to the reader.

Results

Please add a table about the research participants, including demographic details and details related to their bereavement (type of loss, time that has passed, relationship with the deceased and so on). In presenting the findings, we recommend reducing the direct quotations from the participants and providing more in-depth interpretation and discussion of emotional depth.

Discussion

The discussion covers the main findings. We recommend tying the findings to theories of coping with bereavement and loss and/or psychological explanations for coping with anxiety and stress. There is no discussion of the findings regarding age patterning. Are there differences between grieving for a first-degree relative and grieving for a friend? Dokd describes disenfranchised grief that is liable to make coping more difficult.

Author Response

Perceptions of the use of alcohol and drugs after sudden bereavement by unnatural causes: analysis of online qualitative data

14 January 2019

Response to peer reviewers’ comments:

Reviewer 1

Does the introduction provide sufficient background and include all relevant references? Must be improved Is the research design appropriate? Must be improved Are the methods adequately described? Must be improved Are the results clearly presented? Must be improved Are the conclusions supported by the results? Yes

This wide-ranging qualitative study examined texts written by 363 adolescents after the death of someone close to them. This study is very important and can help in understanding the consequences of bereavement on risky behaviors such as alcohol and drug use. The authors need to reinforce the theoretical anchor of the research, formulate the qualitative research procedure such that it will be clear to the reader and provide a more in-depth analysis of the data. Therefore, I recommend accepting the paper after the following changes have been made:

Introduction

Some references should be replaced with more recent literature. For example: #10

Reference 10 was a 1996 Finnish study (Martikainen P, Valkonen T. Mortality after the death of a spouse: Rates and causes of death in a large Finnish cohort. Am J Public Health. 1996 Aug;86(8 I):1087–93.) and we had selected this because more recent studies investigating mortality risk after bereavement (Boyle et al 2011; Elwert & Christakis, 2008) have not been so specific about alcohol-related causes of death. As this is the only study to show this level of detail we have kept this in, but also added the above two studies to paragraph 1 of the Introduction to reinforce the point about mortality risk using more recent research findings.

The other references in the Introduction pre-dating 2000 are the classic Lancet review from 2007 (Stroebe et al), which has not been updated, and two other important studies from 2003 (Kaltman &, Bonanno), which we are keen to retain.  To provide more contemporary primary research findings we have also added Zisook and Shear, 2013; Shah et al, 2013, and Kristensen et al, 2102 to the Introduction.

The authors did not define complicated grief. 

We have now added to the first paragraph of the Introduction some text to describe complicated grief in the context of the new diagnostic classification systems, and added that there is an increased risk of complicated grief after violent losses (Lobb et al, 2010).

What percentage of the bereaved adolescents use alcohol and drugs in comparison with the general (non-bereaved) adolescent population?

We have added a study by Brent et al, which found that when comparing adolescents bereaved by the death of a parent to the non-bereaved, 4.5% versus 0.0% were identified as having an alcohol or substance use disorder. We did not identify research that has investigated this in adolescents bereaved by the loss of any close contact, but have cited Brent et al 2009 in our introduction as parental (and grandparent) death is probably the most common cause of bereavement in this group.

The introduction surveys research in this field but lacks a theoretical anchor. Many theories have been proposed for coping with grief and loss. I recommend the two-track bereavement theory proposed by Prof. Rubin.

We have added to the Introduction a section providing an overview of theories of bereavement, including Rubin’s two-track bereavement theory. We have also added to the discussion a section in which we relate our sub-themes to this body of theory, to explore whether the patterns described might be considered normal or abnormal grieving. We did not feel comfortable focussing on one theoretical stance, but instead presented the breadth of theory.

Materials and Methods

The authors need to indicate what type of qualitative research they conducted and why: phenomenological? narrative? ethnographic?

This was a mixed methods study in which participants were given the opportunity to expand on their responses to a questionnaire through written text to describe the phenomenon of substance use after bereavement.  We have added to the text that we took a phenomenological approach, albeit with the limited free text provided by respondents online.

The data were collected in 2010. Please add an explanation of the current relevance of the analysis or indicate the research limitations.

We have added to the limitations that although the data were collected a decade ago it still provides valuable insights because Global Burden of Disease data for the period 1990 to 2013 show that alcohol remains the leading risk factor for ill-health, early mortality and disability among people aged 15 to 49 years (Forouzanfar et al, 2015 https://www.thelancet.com/journals/lancet/article/PIIS0140-6736(15)00128-2/fulltext#seccestitle10). It is also the only study that has investigated this research question, so these data constitute the best available evidence.

The writers put together the questionnaire. What sort of validation process was used? Was the questionnaire tested in a pre-test?

We designed the questionnaire in collaboration with a consultation group of young bereaved adults and bereavement counsellors, who identified important domains to cover in relation to the impact of bereavement. It was then piloted with individuals accessing support from four national bereavement support organisations (Cruse Bereavement Care, Samaritans, Survivors of Bereavement by Suicide, and Widowed by Suicide) and amended following that. We have added this information to the Methods.

Appendix S1 wasn’t attached.

This is the survey instrument used and is available to download at this link from another issue of this journal:

https://www.mdpi.com/1660-4601/16/21/4093/s1

Further information about the 'reflection process' (page 4) would make this section more meaningful to the reader.

In this section we had explained that: “Regular discussion meetings within the research team encouraged reflexivity and enhanced the validity of our findings by providing opportunities to challenge others’ assumptions and refine our interpretations and analytic processes”. We have added to this: 

“Senior authors brought perspectives from the disciplines of medical sociology (Fiona Stevenson) and academic psychiatry (Michael King, David Osborn). One aspect of our discussions was our interpretation of the degree to which respondents appeared to feel that that use of substances was harmful, and the extent to which an awareness of harmfulness was balanced with any perceived benefits.”

Results

Please add a table about the research participants, including demographic details and details related to their bereavement (type of loss, time that has passed, relationship with the deceased and so on).

We have added Table 1 to present socio-demographic characteristics of the bereaved, but have excluded the 120 people who filled in free text data only to indicate “not applicable” or equivalent.

This has brought our sample down to 243, and we have amended it to this in the abstract and results.

In presenting the findings, we recommend reducing the direct quotations from the participants and providing more in-depth interpretation and discussion of emotional depth.

We have edited the text to delete specific quotes and to add more interpretive content, particularly in relation to identifying links between sub-themes.

.

Discussion

The discussion covers the main findings. We recommend tying the findings to theories of coping with bereavement and loss and/or psychological explanations for coping with anxiety and stress.

We have added to the Discussion a section in which we relate our sub-themes to the bereavement theories described in the Introduction. This considers the extent to which the patterns described might be considered normal or abnormal grieving.

There is no discussion of the findings regarding age patterning. Are there differences between grieving for a first-degree relative and grieving for a friend? Doka describes disenfranchised grief that is liable to make coping more difficult. 

We had reported our findings regarding age patterning in the abstract and in the main findings of the discussion. We had also mentioned gender patterning in the main findings section. However, we have added to this that we did not find any differences between grieving for a first-degree relative and grieving for a friend. There were no apparent examples of the ‘hidden bereaved’ and so we did not discuss the concept of disenfranchised grief for reasons of space.

Reviewer 2 Report

This paper addresses an important and often underresearched topic. It is well written and was interesting to read. Just a few comments:

I appreciate that the researchers mention their own positionality in regards to the subject matter, although their attitudes could be explained in a little more detail in the text.

Given that the sample was recruited through higher education institutions, these participants could be different from the general population. This needs to be acknowledged upfront, for example in the abstract.

In the section named Analytic approach, I think it is unnecessary to name the authors in the text.

More details need to be provided on the type of coding used, I would recommend Saldana´s book on qualitative coding to name particular strategies used. In addition, the steps taken in thematic analysis also need to be outlined.

The last few lines of the method seem strange, are they correct?

Line 189, what is FTS?

It is good to give contextual information after each quote, but they all need to be in a similar format, for example in lines 202-203 it is stated that the person was 17 years old when the friend died. It would be helpful just give the number of years since the event, not the age.

Is there any information available on what distinguishes the different themes? They seem to be in contrast with each other, are there any factors that could explain this difference?

Saldaña, J. (2015). The coding manual for qualitative researchers. Sage.

Round 2

Reviewer 1 Report

The paper has been revised following the comments and suggestions of the reviewer.